# The Insignificant Correlation between Androgen Deprivation Therapy and Incidence of Dementia Using an Extension Survival Cox Hazard Model and Propensity-Score Matching Analysis in a Retrospective, Population-Based Prostate Cancer Registry

**DOI:** 10.3390/cancers14112705

**Published:** 2022-05-30

**Authors:** Young Ae Kim, Su-Hyun Kim, Jae Young Joung, Min Soo Yang, Joung Hwan Back, Sung Han Kim

**Affiliations:** 1National Cancer Control Institute, National Cancer Center, Goyang 10408, Korea; 12274@ncc.re.kr (Y.A.K.); ms1230@ncc.re.kr (M.S.Y.); 2Department of Neurology, Research Institute and Hospital of National Cancer Center, Goyang 10408, Korea; herena20@ncc.re.kr; 3Department of Urology, Urological Cancer Center, Research Institute and Hospital of National Cancer Center, Goyang 10408, Korea; urojy@ncc.re.kr; 4Health Insurance Policy Research Institute, National Health Insurance Service, Wonju 26464, Korea; 25125@ncc.re.kr

**Keywords:** dementia, incidence, androgen, prostate cancer, extension survival

## Abstract

**Simple Summary:**

This study shows the insignificant effect of the duration of androgen-deprivation therapy on the incidence of dementia in patients with prostate cancer from population-based data. We found that, despite an overall lower incidence of dementia in the androgen-deprivation-therapy group compared to the non-therapy group, there were no significant correlations between androgen-deprivation therapy and the prevalence of individual dementia subtypes in patients with prostate cancer. This demonstrates that patients with old age, obesity, regional SEER stage, a history of cerebrovascular disease, and a high Charlson Comorbidity Index were at increased risk for dementia.

**Abstract:**

This study aims to evaluate the effect of androgen-deprivation therapy (ADT) on the incidence of dementia, after considering the time-dependent survival in patients with prostate cancer (PC) using a Korean population-based cancer registry database. After excluding patients with cerebrovascular disease and dementia before or within the 3-month-ADT and those with surgical castration, 9880 (19.3%) patients were matched into ADT and non-ADT groups using propensity-score matching (PSM) among 51,206 patients registered between 2006 and 2013. To define the significant relationship between ADT duration and the incidence of dementia, the extension Cox proportional hazard model was used with *p*-values < 0.05 regarded as statistically significant. The mean age and survival time were 67.3 years and 4.33 (standard deviation [SD] 2.16) years, respectively. A total of 2945 (9.3%) patients developed dementia during the study period, including Parkinson’s (11.0%), Alzheimer’s (42.6%), vascular (18.2%), and other types of dementia (28.2%). Despite PSM, the PC-treatment subtypes, survival rate, and incidence of dementia significantly differed between the ADT and non-ADT groups (*p* < 0.05), whereas the rate of each dementia subtype did not significantly differ (*p* = 0.069). A multivariate analysis for dementia incidence showed no significance of ADT type or use duration among patients with PC (*p* > 0.05), whereas old age, obesity, regional SEER stage, a history of cerebrovascular disease, and a high Charlson Comorbidity Index were significant factors for dementia (*p* < 0.05). Insignificant correlation was observed between ADT and the incidence of dementia based on the extension survival model with PSM among patients with PC.

## 1. Introduction

Worldwide, newly diagnosed patients with prostate cancer (PC) starting androgen-deprivation therapy (ADT) were estimated to be approximately 50% of all PC patients (25,000 patients annually) [1,2]. The incidence and types of ADT-related complications have also been found to increase proportionally to the exposure time to ADT, since ADT use has become the standard treatment for both androgen-dependent and androgen-independent PC, and its duration has increased from 12–24 months in the past to 3–5 years currently with other new agents added [2,3,4,5].

A recently emerging and highly debated issue on ADT-related adverse events is the possibility of impaired cognitive dysfunction, such as dementia [6], due to the irreversible, multifactorial, detrimental, exposure time-dependent, and general performance status-dependent impacts of ADT on the central nervous system, shown in both preclinical ex vivo and in vivo studies [7,8,9] and clinical non-randomized studies, especially for the Alzheimer’s dementia [6,10]. Without large-scale, long-term follow-up randomized controlled studies, the association between ADT and dementia incidence cannot be determined, although some multicentric/population-based studies or baseline controlled studies with different groups using matching methodology could be utilized to draw a conclusion [8,10,11].

As the prognostic outcome of PC and the incidence of dementia associated with ADT exposure are time-dependent factors [12], it is important to consider time-dependent survival variables in a multivariate analysis when investigating ADT as a risk factor for survival and dementia prevalence. Indeed, these survival variables might have had an influence on the contradictory inferences obtained by previous studies [8,10,11,12,13]. Therefore, using data from a Korean population-based cancer registry, this study aimed to evaluate the effect of ADT on the incidence of dementia in patients with PC, after considering time-dependent extension survival methodology and after eliminating the influences of inherent selection biases, by applying an extension survival model and propensity-score matching (PSM) on PC cohorts with and without ADT use.

## 2. Materials and Methods

### 2.1. Population-Based Cancer Registry Database

The analytic methodology for the anonymized the Korean National Health Insurance System of Statistics (NHIS) database has been described in detail in our previous epidemiological articles [4,13]. This study used cohorts with cancer incidence and mortality data from 2006 to 2013 in the NHIS combined Korean Central Cancer Registry, in which accurate recording of cancer registration data followed a standardized cancer diagnosis dating system. Hence, the possibility of multiple diagnosis dates in previously diagnosed patients was excluded, and only accurately classified existing and new patients were included.

### 2.2. Study Sample

A total of 51,206 patients with PC between 2006 and 2013 were included based on the International Classification of Diseases, Tenth Edition (ICD-10) diagnosis code C61, and followed-up until the end of 2014. Patients with a history of dementia or cerebrovascular disease prior to PC diagnosis (N = 15720, 30.7%) were excluded. Patients within 3-months of PC diagnosis (N = 1039, 2.0%); within 3-months of initiating ADT (N = 372, 0.7%); with less than 3-months of ADT history (N = 2179, 4.3%); and a history of surgical castration (N = 325, 0.6%) for a total of 34,075 patients with complete medical records, including 16,827 (49.4%) patients treated with ADT, were included in the study.

During the study period, a total of 2945 (8.60%) patients with PC with a newly developed diagnosis of dementia were enrolled, based on the ICD-9 and ICD-10 codes for dementia and cognitive dysfunction (F00~03 and G309). The dementia subtypes were classified into Alzheimer’s dementia (F00 or G309), vascular dementia (F01), Parkinson’s dementia (F02), other non-Parkinson’s dementia (F02), and unspecified dementia (F03). Furthermore, patients were enrolled only if they had a longer than 3-month history of dementia medication use and had been diagnosed more than twice within 6 months following the initial diagnosis by a neurologist, to exclude elderly patients taking prophylactic dementia medications without a diagnosis of dementia.

Other clinico-pathological parameters including underlying disease, body mass index (BMI), Charlson Comorbidity Index (CCI), and medication use were selected according to the methods used in our previous epidemiological studies [4,14]. Only patients using statin and aspirin as preventive medical therapy for cardiovascular disease were considered. SEER (Surveillance, Epidemiology, and End Results) stage classification was used to determine the stage of PC (local, regional, distant, and unknown). The primary therapeutic strategies for PC treatment included surgery, radiation, and chemotherapy, which were coded under EDI-CD.

During the study period, a total of 84 hormonal agents approved by the Korean FDA, including gonadotrophin-releasing hormone (GnRH) agents (goserelin, leuprorelin, and triptorelin acetate) and anti-androgen agents (flutamide and bicalutamide), were used for ADT. The finally enrolled ADT group comprised patients treated with ADT for at least 3 months.

### 2.3. Propensity-Score Matching Analysis

PSM of 1:1 was performed between the ADT and non-ADT groups to prevent significant baseline differences and resulted in a final enrollment of 9880 patients. To characterize the insignificant differences between ADT and non-ADT groups in PC, a 1249-paired PSM 1:1 was performed. The matched variables, which were significantly different between ADT and non-ADT groups, were age at cancer diagnosis, BMI, SEER stage, a history of smoking, underlying disease (hypertension, diabetes mellitus, and cardiovascular disease), CCI, and the medication (aspirin and statin) used at the time of the study period. 

After matching, a test for homogeneity between the groups was performed using a chi-squared or Mc Nemar’s test for categorical data, a paired t-test for continuous data, and analysis of variance (ANOVA) to explain the effect of each confounding variable on the incidence of dementia. Lastly, a PSM of 1:1 was performed according to the disease states to analyze the relationship between the use of ADT and the incidence of dementia. To adjust for confounding variables, we performed regression analysis to test the difference in the resulting variables.

### 2.4. Extended Model Analysis

An extended survival model [9] was developed for the duration and type of the ADT variables to control for treatment duration and type that affects time-dependent outcome variable. The survival intervals were divided into 30 days and included in the survival period of each group according to the duration and type of the ADT at each survival interval.

### 2.5. Multivariate Cox Analysis

The paired t-test and Chi-square or Mc Nemar’s test were used to determine the differences between groups. A multivariate Cox analysis was used to define the significant effect of ADT and its duration on dementia incidence during the study period. From the starting date of ADT, hazard ratios (HRs) and 95% confidence interval (CI) values for the occurrence of dementia and death events among the ADT groups were analyzed using Cox proportional hazard models. 

To adjust the Cox proportional models, age, BMI, a history of smoking, SEER stage, medication (aspirin and statin), underlying diseases (diabetes, hypertension, and other cardiovascular diseases), and the CCI were used. Two-sided *p*-values < 0.05 were considered statistically significant. All statistical analyses were performed using SAS (release 9.4, SAS Institute Inc., Cary, NC, USA).

## 3. Results

### 3.1. Overall Patients’ Baseline Characteristics

The mean age, ADT duration, and survival time of the 9880 included patients were 67.3 (SD 6.3) years-old, 960.8 (range, 736.1–741.0) days, and 4.33 (SD 2.16) years, respectively, with a 3.53% of overall death rate, including 2.0% of PC-specific deaths during a mean follow-up time of 4.3 (SD 2.1) years (Table 1). Among 592 (6.0%) patients diagnosed with dementia during the study period, Alzheimer’s dementia (N = 252, 42.6%) was the most prevalent subtype, followed by vascular dementia (N = 108, 18.2%), Parkinson’s disease (N = 65, 11.0%), and other non-Parkinson’s dementia (N = 167, 28.2%). Other clinic-pathological information, including SEER stage and therapeutic modalities, are described in Table 1.

### 3.2. Comparative Results between Matched Cohort of ADT and Non-ADT Groups

The 1:1 matched comparative results between ADT and non-ADT groups demonstrated that the overall dementia incidence was significantly different (ADT, N = 270 [45.6%] vs. non-ADT, N = 322 [54.4%], *p* = 0.025), whereas the incidences of each individual dementia subtype were not different between groups (*p* = 0.069), in which the vascular dementia rate was higher in the non-ADT group (21.74%) than the ADT group (14.07%) (Table 1). 

Although there were no baseline differences in SEER stage, BMI, diagnosed year of cancer, year of dementia diagnosis, underlying diseases, medication, and smoking history between the ADT and non-ADT groups after PSM, there were still significant baseline differences, with significantly higher rates in the ADT group of cancer treatment within 6 months from cancer diagnosis (83.9% vs. 49.6% in non-ADT), chemotherapy (42.0% vs. 10.6%), radiation therapy (16.5% vs. 5.0%), and death (5.4% vs. 1.7%), including PC-specific death (65.7% vs. 15.9%) than in the non-ADT group, which had a significantly higher rate of surgical treatment (46.8%) than the ADT group (28.9%) (*p* < 0.05, Table 1).

### 3.3. Multivariate Results of Dementia Incidence

In the multivariate analysis among the 1:1 PS-matched patients (total 9880) with PC, ADT duration and ADT therapeutic types were not significant risk factors for dementia incidence (*p* > 0.05), whereas old age (HR > 1.0), obese BMI (25~30, HR 0.676, CI 0.545–0.838), regional SEER stage (HR 0.720, CI 0.578–0.897), presence of cardiovascular disease (HR 3.083, CI 2.543–3.737), and high CCI (HR 1.479 for CCI 1 and HR 1.961 for CCI2+) were significant risk factors for dementia (*p* < 0.05, Table 2).

### 3.4. Comparative Results between Dementia and Non-Dementia Groups

In the comparison between 1:1 PS-matched groups (2498 patients) with dementia and without dementia by SEER stage, BMI, diagnosed year of cancer, year of dementia diagnosis, underlying diseases, medication, and smoking history (Table 3), the dementia group had a significantly longer interval between cancer diagnosis and treatment initiation within 6 months (59.8% vs. 38.35%), and lower rates of primary treatment, such as surgery/chemotherapy/radiation (1.0/26.2/2.7% vs. 26.2/27.0/3.7%), higher rate of death (12.4% vs. 7.5%), shorter duration of ADT use (843.9 vs. 1243.0 days) and shorter time interval from cancer diagnosis to ADT initiation (89.4 vs. 143.2 days) (*p* < 0.05).

### 3.5. Multivariate Results of Dementia Incidence According to SEER Tumor Staging

In the stratified comparison according to SEER staging, Table 4 and Table 5 show the multivariate analysis of risk factors for dementia among local/regional and distant/unknown SEER stage patients after 1:1 PSM. The results showed that old age (HR 1.965–2.860), obesity BMI (HR0.711, CI 0.564–0.896), high CCI (HR 1.1468–1.716), and ADT duration between 3–24 months (HR 0.224-0.410) were significant risk factors for dementia among local/regional SEER staged patients (*p* < 0.05, Table 4); age > 75 year-old (HR 7.995, CI 2.159–29.604), underweight BMI (HR 3.425, CI 1.421–8.258), and high CCI (HR 1.919 for CCI 1 and HR 2.201 for CC2+) were significant risk factors among distant/unknown SEER stage patients (*p* < 0.05, Table 5).

## 4. Discussion

ADT is a standardized therapy for PC, and it blocks the androgen effects resulting in a rapid drop of testosterone concentration, leading to andropause to prevent from PC progression. However, ADT had a detrimental exposure-dependent effect on general performance status, including impaired cognitive dysfunction, such as the development of dementia [15]. Some contradictory results denied the insignificantly direct correlation of ADT exposure time to dementia incidence in PC [16,17,18]. 

This study used an extended survival methodology to evaluate the time-dependent influences of dementia by the use of ADT and also concluded that a significantly direct relationship between ADT and its duration and dementia incidence have not been shown, especially for Alzheimer’s dementia in a Korean population cohort. This study considered the time-dependent effect of extended exposure of ADT on dementia and disease burden states by the categorized SEER staging. 

We applied strict diagnostic indications of dementia, and eliminated the effect of the inherent bias of baseline differences between ADT and non-ADT cohorts by PSM, even after considering all the adjusted major known etiologies of cognitive dysfunction and all the disease burden states according to SEER staging, especially in an Asian population-based cancer registry [7,16,19,20,21,22,23]. In the absence of randomized clinical trials, our analytic method is the best approach to demonstrate the insignificant relationship between ADT exposure and dementia in real-world clinical settings.

Contrary to previous studies showing a significant direct relationship between ADT and dementia incidence, the insignificant relationship between ADT exposure and the incidence of dementia we observed in this study might be explained by multiple aspects, such as the different characteristics of the cohorts, geographical prevalence of dementia in different ethnical backgrounds, and their enrollment criteria. This study applied different selection criteria of dementia compared to other similar population-based studies [15,16,17]. We enrolled patients with a more than 3-month prescription history of dementia medication and who were diagnosed more than twice within at least 6 months from the first diagnosis. Those who were taking prophylactic dementia medications were excluded from the analysis.

Other reasons for the insignificant results could be the different ethnicity and geographical locations of the cohorts [7,15,21]. Hence, the study cohorts likely differed in terms of immunological profiles [8], social background and lifestyles [18,24], the regional prevalence rate of dementia [25,26], and different ADT-dosing policies based on ethnicity-related anthropometric characteristics and androgen-physiologic profiles [27]. Previous studies using Asian cohorts showed a similarly insignificant relationship between ADT and dementia prevalence, especially with ADT-related Alzheimer’s disease. Asians reportedly have a lower rate of Alzheimer’s dementia and higher rate of non-Alzheimer’s disease, such as vascular dementia, compared with those seen in Caucasians and African Americans [24,28,29] due to different familial and social lifestyles, dietary habits, lifespans, and genetic/immunological backgrounds [25,26,27].

Koreans are part of the East Asian population, which includes the populations of Japan, China, and Taiwan. These countries culturally possess strong family bonds based on the Han-ethnic and Confucianism culture, a highly hierarchic male-dominant social lifestyle with higher socioeconomic burden in males, geographically overcrowded urban lifestyle [25], and a nationally obligatory Bacille Calmette–Guerin vaccination program [9], which could account for the reduced risks of Alzheimer’s dementia but increased risks of non-Alzheimer’s dementia, such as vascular dementia [24,29]. This study also showed an insignificant difference in the rate of Alzheimer’s dementia between the ADT (42.96%) and non-ADT (42.24%) groups, whereas a higher rate of vascular dementia was found in the non-ADT group (14.07% ADT vs. 21.74% non-ADT, Table 1) [16].

This study compared the dementia and non-dementia groups to identify any differential characteristics among the patients (Table 3), revealing the significant importance of tumor characteristics and their relationship to the patients’ general performance status in PC cohorts, which affects the dementia incidence. Since the patients’ general condition significantly affected the incidence of dementia, the tumor burden and extent were important factors for not only general performance and nutritional state but also mental health, including cognitive function [7,19,20,30]. 

The dementia group had a significantly poorer general condition (underweight BMI, higher CCI, and higher rates of hypertension/diabetes mellitus/aspirin use), higher tumor extent (higher rate of distant staging and lower rate of regional staging, and higher rate of early initiation of cancer treatment), and higher rates of no ADT treatment and death. They were also older compared to the non-dementia group. Therefore, a thorough dementia screening program in collaboration with other specialists, including neurologists, nutritionists, psychiatrists, and rehabilitation medicine doctors, should be considered for high-risk PC patients based on these factors.

Other reasons for the contradictory results compared to previous studies could be the ADT duration and study period [9,16,20]. The study period was between 2006 and 2013, during which time most of the patients had ADT histories of less than 3 years, since ADT had been approved by the national insurance system as a first-line treatment only for advanced and metastatic PC cases until biochemical recurrence. After that point, second-line systemic therapies were started without ADT due to the transformation of hormone sensitive PC to hormone-resistant, hormone refractory PC. 

Currently, ADT has become the mainstay treatment and is continuously used in patients with PC until either death or remission, with the newly introduced secondary and chemotherapeutic agents offering prolonged overall survival [12,15]. Thus the number of patients using ADT > 3 years has increased, suggesting there might be a significant relationship between ADT and dementia incidence in this group. Comparable results have been shown by previous studies comparing the effects of similar ethnicities [22,27], dietary lifestyle [31], and psycho-social environments [3] on the incidence of dementia. 

The multivariate analyses in Table 4 and Table 5 showed that ADT was a significant risk factor of dementia only for the localized/regional SEER staging groups, whereas it was not for the distant/unknown SEER staging cohorts. Further enrollment of localized/regional SEER staged patients with PC might reveal a significant relationship between ADT and dementia incidence, because the survival time of this cohort have been prolonged more recently with new therapeutic modalities.

Despite the fact that PSM methodology and extended survival analysis were used to adjust for inherent influencing parameters on dementia incidence, this study had certain limitations: it was a retrospective study that utilized health-claims data, and not all variables influencing dementia, such as exercise, severity and extent of cardiovascular disease, bisphosphonate, follow-up duration, and systemic therapies, such as chemotherapy and anticoagulant agents (cilostazol or clopidogrel), were included in the multivariate analysis. In particular, the significant cardiovascular disease for risk factors of the dementia in this study has also been a well-known factor for both ADT and dementia [19,20]. 

Further concrete evaluation of the severity and extent of cardiovascular disease might help to eliminate the confounding effect of the cardiovascular disease on the cognitive dysfunction. In addition, a strict definition of dementia used in this study and shorter follow-up duration compared with the survival time for PC might eliminate all the possible minor dementia in the analysis as well as elimination of follow-up duration, radiation, and chemotherapy as matching variables. Further study with more concrete adjusting the extent of cardiovascular disease should be considered. However, this was the first population-based retrospective study to use two statistical modalities to control the inherent cohort bias and time-dependent bias to demonstrate the relationship between ADT duration and type and dementia incidence in a PC cohort. Further, current second generation of hormonal agents with ADT should also be considered in future studies.

## 5. Conclusions

This study showed an insignificant correlation between ADT and the incidence of dementia using an extension survival Cox hazard model and PSM analyses. Further prospective trials with large-sized cohorts should be performed to evaluate the true effect of ADT on dementia incidence in PC during the current new therapeutic era of PC.

## Figures and Tables

**Table 1 cancers-14-02705-t001:** Comparison of demographics between ADT and non-ADT groups among patients with prostate cancer after 1:1 propensity score matching adjusted variables for age, BMI, smoking, diagnosed year of cancer and dementia, SEER stage, underlying diseases, medications, and Charlson comorbidity index.

	Total (N = 9880)		ADT_Group (N = 4940)		Non-ADT Group (N = 4940)		*p*-Value
	n	%	N	%	n	%	
**Age group (year-old)**	9880		4940		4940		
50–64	3037	73.73	1554	31.46	1483	68.54	0.1254
65–74	5770	0.22	2875	58.20	2895	41.80	
75-	1073	26.05	511	10.34	562	89.66	
**BMI (missing = 4084) (kg/cm^2^)**							
Underweight (<18.5)	221	2.24	113	2.29	108	2.19	0.9516
Normal (18.5~22.9)	3094	31.31	1548	31.34	1546	31.30	
Overweight (23~24.9)	2886	29.21	1427	28.89	1459	29.53	
Obesity (25~29.9)	3449	34.91	1738	35.18	1711	34.63	
Severe obesity. (30~)	230	2.33	114	2.30	116	2.35	
**Smoking (Missing = 11,099)**							
Ex-smoker	878	8.89	439	8.89	439	8.89	1
current smoker	1234	12.49	617	12.49	617	12.49	
non smoker	7768	78.62	3884	78.62	3884	78.62	
**Year of cancer diagnosis**							
2006–2009	3764	38.10	1886	38.18	1878	38.02	0.8684
2010–2013	6116	61.90	3054	61.82	3062	61.98	
**SEER**							
Local	5982	60.55	2991	60.55	2991	60.55	1
Regional	2842	28.77	1421	28.77	1421	28.77	
Distant	44	0.44	22	0.45	22	0.44	
Unknown	1012	10.24	506	10.24	506	10.24	
**Interval time between cancer diagnosis and treatment initiation within 6 months**	6596	66.76	4144	83.89	2452	49.64	<0.0001
**Underlying disease**							
Hypertension	4900	49.60	2450	49.60	2450	49.60	1
Diabetes	2550	25.81	1275	25.81	1275	25.81	1
Cardiovascular disease	920	9.31	460	9.31	460	9.31	1
**Medication**							
Aspirin	3802	38.48	1901	38.48	1901	38.48	1
Statin	2388	24.17	1194	24.17	1194	24.17	1
**Treatment type**							
Surgical Treatment	3737	37.82	1427	28.89	2310	46.76	<0.0001
Chemotherapy	2650	26.82	2108	42.67	542	10.97	<0.0001
Radiation	1075	10.88	845	17.11	230	4.66	<0.0001
No treatment	2418	24.48	560	11.33	1858	37.61	<0.0001
**Pathology**							
adenocarcinoma	8220	83.20	4086	82.71	4134	83.68	0.1965
no adeno	1660	16.80	854	17.29	806	16.32	
**Charlson comorbidity index**							
0	4348	44.01	2174	44.01	2174	44.01	1
1	4266	43.18	2133	43.18	2133	43.18	
2+	1266	12.81	633	12.81	633	12.81	
**Survival(death)**	349	3.53	265	5.36	84	1.70	<0.0001
**Cancer death**							
Prostate cancer	187	53.58	174	65.66	13	15.48	<0.0001
other cancer	39	11.17	17	6.42	22	26.19	
Other	123	35.25	74	27.92	49	58.33	
**Survival time(year, mean [SD])**	4.33 (2.16)		4.31 (2.15)		4.34 (2.16)		0.5926
**Dementia**	592	100	270	45.6	322	54.4	0.025
Parkinson	65	10.98	30	11.11	35	10.87	0.0691
Alzheimer	252	42.57	116	42.96	136	42.24	
Vascular dementia	108	18.24	38	14.08	70	21.74	
Other dementia	167	28.21	86	31.85	81	25.15	
**ADT treatment**					NA		
only GnRH agonist	340	6.88	344	6.88	-	-	-
only Antiandrogen	706	14.29	760	14.29	-	-	-
Both	3894	78.83	4013	78.83	-	-	-
**Duration of ADT (days, mean (SD))**					NA		
Total	960.81(736.11)		960.81(736.11)				
only Antiandrogen	711.39(617.76)		711.39(617.76)		-		-
only GnRH agonist	692.37(506.89)		692.37(506.89)		-		-
Both	1029.47(757.41)		1029.47(757.41)		-		-
**ADT Duration (days, mean (SD))**							-
No ADT	-		-		-		-
3 ≤ ADT < 6	130.45(28.90)		130.45(28.90)		-		-
6 ≤ ADT < 12	268.82(53.84)		268.82(53.84)		-		-
12 ≤ ADT < 24	528.75(104.34)		528.75(104.34)		-		-
24 ≤ ADT	1486.90(626.09)		1486.90(626.09)		-		-
**Interval time from cancer diagnosis to ADT start(days, mean(SD), median)**					NA		
Total	186.45(370.44)		186.45(370.44)				
only Antiandrogen	319.47(496.64)		319.47(496.64)		-		-
only GnRH agonist	148.66(321.02)		148.66(321.02)		-		-
Both	343.14 (474.36)		343.14 (474.36)		-		-

NA, not applicable; ADT, androgen-deprivation therapy; SEER, Surveillance, Epidemiology, and End Results; SD, standard deviation; GnRH, Gonadotrophin-releasing hormone; and BMI, body mass index.

**Table 2 cancers-14-02705-t002:** Multivariate cox regression of risk factors for dementia among overall patients after propensity-score matching and extended survival analysis.

	Univariate				Multivariate			
Characteristics	HR	95% CI		*p*-Value	aHR	95% CI		*p*-Value
**Age group (year-old)**								
50 ≤ Age < 65	1				1			
65 ≤ Age ≤ 75	2.103	1.686	2.622	<0.0001	1.797	1.426	2.265	<0.0001
75 < Age	3.144	2.389	4.137	<0.0001	2.392	1.781	3.213	<0.0001
**BMI (kg/cm^2^)**								
Normal (18.5~22.9)	1				1			
Underweight (<18.5)	1.453	0.946	2.232	0.0877	1.235	0.786	1.939	0.3598
Overweight (23~24.9)	0.822	0.675	1.002	0.0522	0.886	0.722	1.086	0.2431
Obesity (25~)	0.617	0.504	0.756	<0.0001	0.676	0.545	0.838	0.0004
Severe obesity. (30~)	0.585	0.311	1.103	0.0975	0.655	0.346	1.241	0.1943
**Smoking**								
non smoker	1							
current smoker	0.928	0.723	1.19	0.5548				
ex smoker	0.73	0.529	1.008	0.0562				
**SEER**								
Local	1				1			
Regional	0.635	0.517	0.780	<0.0001	0.720	0.578	0.897	0.0033
Distant	1.822	0.754	4.401	0.1824	1.938	0.799	4.704	0.1434
Unknown	1.190	0.933	1.517	0.1613	1.145	0.889	1.475	0.2939
**Medication**								
Aspirin (Ref. = no)	1.224	1.041	1.441	0.0147	1.010	0.824	1.238	0.9255
Statin (Ref. = no)	0.979	0.810	1.183	0.6018				
**Underlying disease**								
CVD	3.423	2.843	4.122	<0.0001	3.083	2.543	3.737	<0.0001
Hypertension (Ref. = no)	1.236	1.051	1.452	0.0104	1.058	0.861	1.299	0.5937
Diabetes mellitus (Ref. = no)	1.230	1.031	1467	0.0214	0.902	0.731	1.114	0.3387
**Charlson comorbidity index**								
0	1				1			
1	1.550	1.290	1.862	<0.0001	1.479	1.217	1.798	<0.0001
2+	2.180	1.732	2.745	<0.0001	1.961	1.510	2.547	<0.0001
**ADT type**								
No ADT	1				1			
only Antiandrogen	0.738	0.517	1.052	0.0931	0.620	0.291	1.319	0.2144
only GnRH agonist	1.075	0.703	1.643	0.7385	0.893	0.393	2.028	0.7865
Both	1.058	0.885	1.264	0.5361	0.812	0.396	1.666	0.5701
**ADT Duration(month)**								
No ADT	1				1			
3 ≤ ADT < 6	0.649	0.419	1.008	0.0541	0.766	0.334	1.759	0.5302
6 ≤ ADT < 12	0.957	0.7	1.308	0.7807	1.193	0.551	2.583	0.6539
12 ≤ ADT < 24	0.761	0.575	1.006	0.0548	0.922	0.432	1.969	0.8345
24 ≤ ADT	1.595	1.274	1.997	<0.0001	1.847	0.881	3.873	0.1045

NA, not applicable; ADT, androgen-deprivation therapy; SEER, Surveillance, Epidemiology, and End Results; SD, standard deviation; GnRH, Gonadotrophin-releasing hormone; and BMI, body mass index.

**Table 3 cancers-14-02705-t003:** Comparison of the demographic characteristics between dementia and non-dementia groups among patients with prostate cancer after 1:1 propensity score matching adjusted variables for age, SEER stage, diagnosed year, aspirin use, statin use, BMI, smoking, hypertension, diabetes mellitus, and cardiovascular disease.

	Total (N = 2498)		Dementia Group (N = 1249)		Non-Dementia Group (N = 1249)		*p*-Value
	n	%	n	%	n	%	
**Age group (year-old)**							
50–64	375	15.01	190	15.21	185	14.81	0.5583
65–74	1412	56.53	693	65.44	719	57.57	
75-	711	28.46	366	89.71	345	27.62	
**BMI (missing = 4084) (kg/cm^2^)**							
Underweight (<18.5)	83	3.32	42	3.36	41	3.28	0.9968
Normal (18.5~22.9)	931	37.27	468	37.47	463	37.07	
Overweight (23~24.9)	721	28.86	362	28.98	359	28.74	
Obesity (25~)	722	28.90	357	28.58	365	29.22	
Severe obesity (30~)	41	1.64	20	1.60	21	1.68	
**Year of cancer diagnosis**							
2006–2009	1359	54.40	699	55.96	660	52.84	0.1172
2010–2013	1139	45.60	550	44.04	589	47.16	
**Smoking (Missing = 11,341)**							
Ex-smoker	196	7.85	98	7.85	98	7.85	1
current smoker	350	14.01	175	14.01	175	14.01	
non smoker	1952	78.14	976	78.14	976	78.14	
**SEER**							
Local	1468	69.57	734	58.77	734	58.77	1
Regional	504	23.89	252	20.18	252	20.18	
Distant	138	6.54	69	5.52	69	5.52	
Unknown	388	18.39	194	15.53	194	15.53	
**Interval time between cancer diagnosis and treatment initiation within 6 months**	1226	49.08	747	59.81	479	38.35	<0.0001
**Underlying disease**							
Hypertension	1380	62.67	690	62.67	690	62.67	1
Diabetes	822	37.33	411	37.33	411	37.33	1
**Medication**							
Aspirin	1114	44.60	557	44.60	557	44.60	1
no Aspirin	1384	55.40	692	55.40	692	55.40	
Statin	636	25.46	318	25.46	318	25.46	1
no Statin	1862	74.54	931	74.54	931	74.54	
**Treatment type**							
Surgical Treatment	437	16.96	9	0.95	428	26.23	<0.0001
Chemotherapy	688	26.71	247	26.17	441	27.02	<0.0001
Radiation	85	3.30	25	2.65	60	3.68	0.0001
Hormonal therapy	1366	53.03	663	70.23	703	43.08	0.1079
**Pathology**							
adenocarcinoma	1050	82.07	1037	83.03	1013	81.10	0.2107
no adenocarcinoma	448	17.93	212	16.97	236	18.90	
**Charlson comorbidity index**							
0	824	32.99	412	32.99	412	32.99	1
1	1154	46.20	577	46.20	577	46.20	
2+	520	20.82	260	20.82	260	20.82	
**Survival(death)**	258	10.33	161	12.44	97	7.50	<0.0001
**Cancer death**							
Prostate cancer	164	63.57	95	59.01	69	71.13	0.1426
other cancer	22	8.53	15	9.32	7	7.22	
Other	72	27.91	51	31.68	21	21.65	
**Survival year (mean, SD)**	5.06(2.75)		5.11(2.10)		5.01(2.05)		0.2034
**Dementia**							
Parkinson	147	11.77	147	11.77	-	-	
Alzheimer	533	42.67	533	42.67	-	-	
Vascular dementia	214	17.13	214	17.13	-	-	
Other dementia	355	28.42	355	28.42	-	-	
**ADT treatment**							
only GnRH agonist	138	5.52	65	5.20	73	5.84	0.2844
only Antiandrogen	83	3.32	45	3.60	38	3.04	
Both	1145	45.84	553	44.28	592	47.40	
No ADT	1132	45.32	586	46.92	546	43.71	
**Duration of ADT (days, mean (SD))**							
Total	1049.28(756.35)		843.86(613.88)		1243.00(824.34)		<0.0001
only Antiandrogen	735.86(625.31)		640.72(542.19)		820.56(683.56)		0.0876
only GnRH agonist	849.06(622.95)		711.06(485.17)		1012.47(727.69)		0.0332
Both	1101.57(768.40)		878.55(625.77)		1309.89(828.94)		<0.0001
**ADT duration (days, mean (SD))**							
No treatment	-		-		-		-
3–6 months	125.73 (28.77)		126.25(27.22)		124.89(31.53)		0.8226
6–12 months	270.24 (55.79)		273.85(52.67)		264.96(60.00)		0.2730
12–24 months	531.79 (102.10)		531.79(106.7)		531.79(95.65)		0.9999
≤24 months	1533.30 (628.15)		1337.40(506.5)		1668.20(667.6)		<0.0001
**Interval time from cancer diagnosis to ADT start (days, mean (SD))**							
Total	117.04(293.25)		89.35(227.21)		143.16(342.24)		0.0006
only Antiandrogen	248.36(503.14)		116.00(267.39)		366.21(623.09)		0.0023
only GnRH agonist	261.52(170.00)		117.62(194.10)		232.03(315.42)		0.0564
Both	97.38(254.40)		83.92(224.59)		109.95(279.00)		0.0813

NA, not applicable; ADT, androgen-deprivation therapy; SEER, Surveillance, Epidemiology, and End Results; SD, standard deviation; GnRH, Gonadotrophin-releasing hormone; and BMI, body mass index.

**Table 4 cancers-14-02705-t004:** Multivariate cox regression of risk factors for dementia among local/regional SEER staging patients after propensity-score matching.

	Univariate				Multivariate			
Parameter	Hazard Ratio	95% Hazard Ratio Confidence Limits		Pr > ChiSq	Hazard Ratio	95% Hazard Ratio Confidence Limits		Pr > ChiSq
**Age group (year-old)**								
50–64	ref				ref			
65–74	2.155	1.562	2.973	<0.0001	1.965	1.422	2.717	<0.0001
75-	2.002	1.463	2.74	<0.0001	2.860	1.905	4.293	<0.0001
**BMI (kg/cm^2^)**								
Normal (18.5~22.9)	ref				ref			
Underweight (<18.5)	1.367	0.806	2.319	0.2462	1.306	0.768	2.219	0.3246
Overweight (23~24.9)	0.849	0.676	1.067	0.1601	0.855	0.680	1.075	0.1799
Obesity (25~29.9)	0.72	0.573	0.903	0.0045	0.711	0.564	0.896	0.0039
Severe obesity. (30~)	0.791	0.418	1.498	0.4716	0.769	0.405	1.461	0.4229
**Smoking**								
non smoker	ref							
Ex-smoker	0.721	0.5	1.04	0.797				
current smoker	0.924	0.695	1.229	0.5863				
**Aspirin (ref = no)**	1.307	1.087	1.57	0.0043	1.112	0.894	1.384	0.3403
**Statin (ref = no)**	1.02	0.826	1.259	0.8563				
**HTN (ref = no)**	1.356	1.128	1.631	0.0012	1.178	0.944	1.471	0.1469
**Diabetes (ref = no)**	1.139	0.93	1.396	0.2089				
**Charlson comorbidity index(CCI)**								
0	ref				ref			
1	1.549	1.264	1.899	<0.0001	1.468	1.194	1.804	0.0003
2+	1.849	1.4	2.442	<0.0001	1.716	1.291	2.281	0.0002
**ADT TYPE**								
No treatment	ref							
only Antiandrogen	0.734	0.499	1.078	0.1151				
only GnRH agonist	0.942	0.568	1.562	0.8158				
Both	0.96	0.783	1.178	0.6985				
**ADT duration**								
No treatment	ref				ref			
3–6 months	0.631	0.389	1.024	0.0624	0.388	0.224	0.673	0.0008
6–12 months	1.022	0.733	1.426	0.8974	0.626	0.410	0.955	0.0296
12–24 months	0.735	0.533	1.014	0.0607	0.451	0.298	0.683	0.0002
≤24 months	1.3	0.993	1.702	0.0561	0.803	0.552	1.168	0.2506

**Table 5 cancers-14-02705-t005:** Multivariate cox regression of risk factors for dementia among distant/unknown SEER staging patients after propensity-score matching.

	Univariate				Multivariate			
Parameter	Hazard Ratio	95% Hazard Ratio Confidence Limits		Pr > ChiSq	Hazard Ratio	95% Hazard Ratio Confidence Limits		Pr > ChiSq
**Age group (year-old)**								
50–64	ref				ref			
65–74	4.503	1.36	14.912	0.0138	4.025	1.208	13.409	0.2736
75–	6.779	2.12	21.678	0.0013	7.995	2.159	29.604	0.0013
**BMI (kg/cm^2^)**								
Normal (18.5~22.9)	ref				ref			
Underweight (<18.5)	2.973	1.243	7.11	0.0143	3.425	1.421	8.258	0.0061
Overweight (23~24.9)	0.706	0.392	1.271	0.2455	0.782	0.431	1.421	0.4196
Obesity (25~29.9)	0.985	0.596	1.629	0.954	1.061	0.632	1.781	0.8226
Severe obesity. (30~)	0.413	0.056	3.023	0.3839	0.435	0.059	3.194	0.4129
**Smoking**								
non smoker	ref							
Ex-smoker	0.781	0.247	2.475	0.6748				
current smoker	0.285	0.09	0.904	0.033				
**Aspirin (ref = no)**	1.204	0.778	1.862	0.4051				
**Statin (ref = no)**	1.648	1.028	2.643	0.0381	1.172	0.693	1.984	0.0159
**HTN (ref = no)**	1.771	1.15	2.728	0.0095	1.418	0.864	2.325	0.5847
**Diabetes (ref = no)**	1.596	1.006	2.532	0.047	1.080	0.638	4.272	0.4268
**Charlson comorbidity index(CCI)**								
0	ref				ref			
1	2.094	1.235	3.553	0.0061	1.919	1.112	3.313	0.0193
2+	2.537	1.375	4.68	0.0029	2.201	1.134	4.272	0.0197
**ADT TYPE**								
No treatment	ref							
only Antiandrogen	1.154	0.407	3.276	0.7879				
only GnRH agonist	2.188	0.771	6.213	0.1413				
Both	1.815	1.13	2.915	0.0137				
**ADT duration**								
No treatment	ref				ref			
3–6 months	0.66	0.149	2.914	0.5832	0.300	0.061	1.467	0.1370
6–12 months	0.877	0.355	2.168	0.7765	0.392	0.136	1.132	0.0836
12–24 months	1.388	0.723	2.664	0.3243	0.601	0.256	1.413	0.2433
≤24 months	2.772	1.543	4.98	0.0006	1.277	0.571	2.859	0.5516

## Data Availability

Not applicable.

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
