# Peer review of "The Insignificant Correlation between Androgen Deprivation Therapy and Incidence of Dementia Using an Extension Survival Cox Hazard Model and Propensity-Score Matching Analysis in a Retrospective, Population-Based Prostate Cancer Registry"

_cancers, 2022, doi:10.3390/cancers14112705_

Round 1

Reviewer 1 Report

The current version addressed most of the concerns raised in previous round of review.

Though the impact of ADT on CVD and diabetes are not relevant to the current report, the authors can update with recent references 

For eg. Ref 19- can be replaced with 

  1. Muniyan, S., et al., Cardiovascular risks and toxicity - The Achilles heel of androgen deprivation therapy in prostate cancer patients. Biochim Biophys Acta Rev Cancer, 2020. 1874(1): p. 188383.
  2. Sun, L., et al., Assessment and Management of Cardiovascular Risk Factors Among US  Veterans With Prostate Cancer. JAMA Netw Open, 2021. 4(2): p. e210070.

Author Response

Though the impact of ADT on CVD and diabetes are not relevant to the current report, the authors can update with recent references. For eg. Ref 19- can be replaced with

Muniyan, S., et al., Cardiovascular risks and toxicity - The Achilles heel of androgen deprivation therapy in prostate cancer patients. Biochim Biophys Acta Rev Cancer, 2020. 1874(1): p. 188383.

Sun, L., et al., Assessment and Management of Cardiovascular Risk Factors Among US  Veterans With Prostate Cancer. JAMA Netw Open, 2021. 4(2): p. e210070.

  • Response: We changed the reference #19 to the following article and added some sentences to explain the relations of cardiovascular disease and the dementia with the ADT (page 18 line 232; page 19 lines 272, 303, and 305-310). We also added two new references in the reference list (page 21) .

  1. Muniyan S, Xi L, Datta K, Das A, Teply BA, Batra SK, Kukreja RC. Cardiovascular risks and toxicity - The Achilles heel of androgen deprivation therapy in prostate cancer patients. Biochim Biophys Acta Rev Cancer. 2020 Aug;1874(1):188383
  2. Sun L, Parikh RB, Hubbard RA, Cashy J, Takvorian SU, Vaughn DJ, Robinson KW, Narayan V, Ky B.Assessment and Management of Cardiovascular Risk Factors Among US Veterans With Prostate Cancer. JAMA Netw Open. 2021 Feb 1;4(2):e210070

Reviewer 2 Report

Dear Authors,

Thank you for addressing all my concerns. I do not any further comments. 

Author Response

  • Thank you for addressing all my concerns. I do not any further comments
  • Response: Thank you for the Reviewer#2 taking time for this manuscript

This manuscript is a resubmission of an earlier submission. The following is a list of the peer review reports and author responses from that submission.

Round 1

Reviewer 1 Report

Dear Authors,

I reviewed with interest the paper entitled “The insignificant correlation between androgen deprivation therapy and prevalence of dementia using an extension survival Cox hazard model and propensity-score matching analysis in a retrospective, population-based prostate cancer registry”.

In this nice original article, the authors aimed to evaluate the effect of ADT on the incidence of different type of dementia, after considering time-dependent survival, in patients with PC, using a Korean population-based cancer registry database. They found an insignificant correlation between ADT and prevalence of dementia.

I found the present study interesting and well written - no major concerns with language editing.

It covers an important topic, since dementia as a possible ADT-related adverse event is still a controversial issue in the urological literature.

- The title is clear and descriptive of what authors have explored in their work.

- The Introduction provides a background which is relevant to the study, yet I would suggest expanding on some relevant topic. For instance, ADT is also continued in the castration resistant PC setting, even when other new agents are added (page 2 lines 44-47) (e.g., DOI: 10.3389/fonc.2021.700258).

- The aim of the paper is well stated, as well as Inclusion and Exclusion criteria.

- Figure 1 should be improved. Indeed, I would suggest specifying that you are using number of patients and %, as well as adding headings at each box to make the Figure more clearly readable. Moreover, at the end of the Figure, I would add 2 boxes including numbers and % of ADT and non-ADT patients.

- Results - which are interesting and significant - are clear and not repetitive.

- Methods are clearly described and in enough detail. Statistical assessment is well conducted and the paper results methodologically correct.

- First line of “Study Sample”: the provided percentage (i.e., 3.27%) is referring to what?! it is not clear. According to Figure 1, it should probably be 100%. Please, correct or clarify.

- Discussion is adequately presented, and interpretations and conclusions are well stated and justified by results.

Minor consideration concerning the journal editing format, please:

- provide a “simple summary”

- delete the first 3 lines of the Results section

I have not further comments.

Author Response

1. The title is clear and descriptive of what authors have explored in their work.

Response: Thank you for your comment.

2. The Introduction provides a background which is relevant to the study, yet I would suggest expanding on some relevant topic. For instance, ADT is also continued in the castration resistant PC setting, even when other new agents are added (page 2 lines 44-47) (e.g., DOI: 10.3389/fonc.2021.700258).

Response: We have changed the sentence with a new reference suggested by the reviewer (page 2 line 48-50). We also added a new reference in the manuscript.

Maggi M, Salciccia S, Del Giudice F, Busetto GM, Falagario UG, Carrieri G, Ferro M, Porreca A, Di Pierro GB, Fasulo V, Frantellizzi V, De Vincentis G, De Berardinis E, Sciarra A. A systematic review and meta-analysis of randomized controlled trials with novel hormonal therapies for non-metastatic castration-resistant prostate cancer: An update from mature overall survival data. Front Oncol. 2021, 11, 700258.

3. The aim of the paper is well stated, as well as Inclusion and Exclusion criteria.

Response: Thank you for your comment.

4. Figure 1 should be improved. Indeed, I would suggest specifying that you are using number of patients and %, as well as adding headings at each box to make the Figure more clearly readable. Moreover, at the end of the Figure, I would add 2 boxes including numbers and % of ADT and non-ADT patients.

Response: We added some thorough explanations about the exclusion criteria according to the previous figure 1 in the Methods section (page 2 lines 84-89). Subsequently, we deleted the previous figure 1 and its legend in the manuscript (page 3).

5. Results - which are interesting and significant - are clear and not repetitive.

Response: Thank you for your comment.

6. Methods are clearly described and in enough detail. Statistical assessment is well conducted and the paper results methodologically correct.

Response: Thank you for your comment.

7. First line of “Study Sample”: the provided percentage (i.e., 3.27%) is referring to what?! it is not clear. According to Figure 1, it should probably be 100%. Please, correct or clarify.

Response: We deleted the “3.27%” phrase to clarify the total number of PC patients in the beginning. (page 2 line 82)

8. Discussion is adequately presented, and interpretations and conclusions are well stated and justified by results.

 Response: Thank you for your comment.

9. provide a “simple summary”

Response: We have added a short summary before the Abstract (page 1 lines 15-22).

10. delete the first 3 lines of the Results section

Response: As per your suggestion, we have deleted the first three lines in the Results section (page 4 line 147-149).

Reviewer 2 Report

1) What was the median and range of follow-up times for individuals included to the study? This should be discussed in the context of previously reported time of dementia diagnosis after the initiation of the ADT. Was the follow-up time in this study sufficiently long to avoid bias?

2) Page 3: "Patients were enrolled only if they had a longer than 3-month history of dementia medication use and had been diagnosed more than twice within 6 months following the initial diagnosis by a neurologist, to exclude elderly patients taking prophylactic dementia medications without a diagnosis of dementia"

This approach needs to be further explained. Patients diagnosed with dementia are frequently not pharmacologically treated. This is common in mild cases, but it is not rare situation even in cases of moderate and serious dementia. How many individuals diagnosed with dementia of any type have not been included to the study, because they had received no pharmacotherapy (or pharmacotherapy for only < 3 months)?

This potential source of bias should be discussed. Investigators intended to exclude "elderly patients taking prophylactic dementia medications without a diagnosis of dementia", but the question remains regarding how often these actually happens and whether filtering-out patients using dementia pharmacotherapy for < 3 months does not introduce more substantial bias than keeping a few patients who were using prophylaxis against dementia. 

3) "The Student’s t-test, chi-squared test, and Fisher’s exact test were used to determine the differences between groups."

These statistical tests do not consider pairing between the propensity matched pairs (in this case, paired t-test can be more appropriate for continuous variables and McNemar's test for dichotomous variables).

4) Propensity scoring needs to be described in more detail. How were the propensity scores calculated (logistic regression,...?), how were the pairs matched (exact matching?....nearest neighbor matching...?...). Justification for the selection of specific variables for propensity matching should be provided in the text. 

Table 1: % values need to be checked and possibly recalculated or better organized. In Age group, % values in columns 4 and 6 seem to represent row percentages (they sum to 100 for two groups of patients), but the values in column 2 correspond to neither row nor column percentages. 

5. "Extended survival analysis" is indicated, but there is no specific information on how it was implemented and how it differed from the traditional Cox proportional hazards model. 

6. ADT+ and ADT- groups display significant differences between chemotherapy  and radiotherapy statuses. Both these treatments had been previously associated with risk of dementia (chemotherapy-associated cognitive impairment; chemotherapy associated with decreased risk of Alzheimer disease). As a result, adjusting for chemotherapy and radiotherapy status is warranted. 

7. It appears that matching did not include follow-up times, which can affect comparison between ADT and non-ADT groups presented in Table 1. 

8. Prevalence is mentioned in the text but this is not an appropriate measure of disease occurrence in this type of studies. 

9. Inclusion criteria start with "A total of 51,206 (3.27%) patients with PC between 2006 and 2013 were included" and in the end indicate "patients who died before PC" as an exclusion criterion. This appears to be not consistent. 

Author Response

Responses to Reviewer #2’s Comments

1)That was the median and range of follow-up times for individuals included to the study? This should be discussed in the context of previously reported time of dementia diagnosis after the initiation of the ADT. Was the follow-up time in this study sufficiently long to avoid bias?

Response: The mean follow time was 4.3 (±2.1) years and the shortest follow-up duration was one year among the matched samples of this study. We have now included the mean follow-up time in the Results section (page 4 lines 149-150). Furthermore, we agree with the reviewer’s comment about the short-term of one-year follow-up for evaluating the incidence of dementia. We have added this limitation in the Discussions section (page 19 lines 305-308). 

2) Page 3: "Patients were enrolled only if they had a longer than 3-month history of dementia medication use and had been diagnosed more than twice within 6 months following the initial diagnosis by a neurologist, to exclude elderly patients taking prophylactic dementia medications without a diagnosis of dementia"

This approach needs to be further explained. Patients diagnosed with dementia are frequently not pharmacologically treated. This is common in mild cases, but it is not rare situation even in cases of moderate and serious dementia. How many individuals diagnosed with dementia of any type have not been included to the study, because they had received no pharmacotherapy (or pharmacotherapy for only < 3 months)?

This potential source of bias should be discussed. Investigators intended to exclude "elderly patients taking prophylactic dementia medications without a diagnosis of dementia", but the question remains regarding how often these actually happens and whether filtering-out patients using dementia pharmacotherapy for < 3 months does not introduce more substantial bias than keeping a few patients who were using prophylaxis against dementia. 

RESPONSE: We used this inclusion criteria for the dementia diagnosis because we excluded patients over diagnosed dementia-like disease and specified the clinical dementia as cohort who needed medication. Several years ago, a Korean national screening program for dementia prevention was launched to screen the dementia-mimicking patients with several questionnaires by the paramedics such that an absolute number of dementia patients has increased being in active surveillance but without any treatment. Therefore, in this study, we focused on the effect of ADT on the incidence of clinical dementia implying a specialized clinician prescribing the dementia-specific medication and excluded all the patients under the active surveillance. We have hence added a limitation of this study to eliminate any low-graded dementia by selecting this dementia definition in the analysis (page 19 lines 303-308).  

3) "The Student’s t-test, chi-squared test, and Fisher’s exact test were used to determine the differences between groups."

These statistical tests do not consider pairing between the propensity matched pairs (in this case, paired t-test can be more appropriate for continuous variables and McNemar's test for dichotomous variables).

Response: We have changed the statistical methodologies in the Methods section (page 3 lines 135).

4) Propensity scoring needs to be described in more detail. How were the propensity scores calculated (logistic regression,...?), how were the pairs matched (exact matching?....nearest neighbor matching...?...). Justification for the selection of specific variables for propensity matching should be provided in the text. 

Table 1: % values need to be checked and possibly recalculated or better organized. In Age group, % values in columns 4 and 6 seem to represent row percentages (they sum to 100 for two groups of patients), but the values in column 2 correspond to neither row nor column percentages. 

Response: Thank you for pointing this out. We have corrected the percentages in table 1 (page 6-7). Moreover, we have included some of the statistical information about the propensity-score matching analysis in the Methods section (page 3 lines 120-126).

  1. "Extended survival analysis" is indicated, but there is no specific information on how it was implemented and how it differed from the traditional Cox proportional hazards model. 

Response: We have used the extended survival methodologies, when the outcome variables might be influenced by the time of exposure which affect the cohort differences according to the time variations. Therefore, this extended survival methodology covers more time variances for the time-dependent outcome variable. We have added a reference explaining the extended survival methodologies in the Methods section (page 3 lines 129-131; page 18 lines 222-224).

  1. ADT+ and ADT- groups display significant differences between chemotherapy  and radiotherapy statuses. Both these treatments had been previously associated with risk of dementia (chemotherapy-associated cognitive impairment; chemotherapy associated with decreased risk of Alzheimer disease). As a result, adjusting for chemotherapy and radiotherapy status is warranted. 

Response: Thank you for your insightful comment. We agree with the reviewer’s comment that chemotherapy and radiotherapy might be important influencing factors on dementia because dementia is closely related to the general condition. Patients who underwent chemotherapy and radiotherapy had a tendency to have either higher tumor burden or poor general conditions with higher CCIs. We also tried to use the therapies in the PS matching at first, but it turned out that the CCI, chemotherapy/radiotherapy/ADT were so closely related that we decided not to use chemotherapy and radiotherapy as a matching variable. Additionally, when we used chemotherapy and radiotherapy as matching variables, the final matching cohort showed a deviated cohort with higher tumor stage and higher CCIs. We have included the possible effect of both therapies on the incidence of dementia in the Discussion section (page 19 lines 303-304).

  1. It appears that matching did not include follow-up times, which can affect comparison between ADT and non-ADT groups presented in Table 1. 

Response: We did not match the follow-up variable in the matching.
The follow-up time was limited till the end of 2017 that some follow-up variations might affect the incidence of dementia. We added this limitation in the discussion section (page 19 lines 303)

  1. Prevalence is mentioned in the text but this is not an appropriate measure of disease occurrence in this type of studies. 

Response: We have changed the term “prevalence” to “incidence” in the text (page 1 line 3, 17,29; page 4 lines 158,159,170,186; page 19 lines 269,270).

  1. Inclusion criteria start with "A total of 51,206 (3.27%) patients with PC between 2006 and 2013 were included" and in the end indicate "patients who died before PC" as an exclusion criterion. This appears to be not consistent. 

 Response: We have corrected the exclusion criteria according to the previous figure 1, which was deleted and their contents were described literally in the Methods section (page 2 lines 84-89).

Reviewer 3 Report

The article “The insignificant correlation between androgen deprivation therapy and prevalence of dementia using an extension survival Cox hazard model and propensity-score matching analysis in a retrospective, population-based prostate cancer registry” by Kim et al. describes the effect of androgen deprivation therapy (ADT) on the incidence of dementia in the Korean population. Here the authors analyzed the Korean population-based cancer registry database, which has details for 51,206 patient between 2003 to 2013. The mean age and survival time were 67.3 years and 4.33 (standard deviation [SD] 2.16) years, respectively. Among them , 2945 (9.3%) patients developed dementia during the study period, including Parkinson’s (11.0%), Alzheimer’s (42.6%), vascular (18.2%) and other types of dementia (28.2%). Despite PSM, the Prostate cancer treatment subtypes, survival rate, and incidence of dementia significantly differed between the ADT and non-ADT groups (p<0.05), whereas the rate of each dementia subtype did not significantly differ (p=0.069). Multivariate analysis showed no significance of ADT type or use duration among patients with PC (p>0.05), whereas old age, obesity, regional SEER stage, a history of cerebrovascular disease, and high Charlson Comorbidity Index were significant factors for dementia (p<0.05). Overall, the authors observed an Insignificant correlation was observed between ADT and the incidence of dementia based on the extension survival model with PSM among patients with PC.

The study is interesting; however, at the present form, it raises many questions.  The article and the results are confusing, and Authors should present in a clarified way.

  • As the years are limited and the prostate cancer survival is longer than the analyzed period(the median survival of PCa patients after AT is 10-15 years), the authors should consider extending the period of analysis.
  • There is a clear definition for the cutoff in 2013. Again, neurological disorders (ND) are higher among patients who received second-generation antiandrogens. Considering the year 2013, it is hard to predict whether the study considers enzalutamide or apalutamide treatment. Again, part of ND is due to the antiandrogen itself and its action on androgen receptors present in neuronal cells.
  • The results are too brief, and the authors should elaborate the findings
  • Did the missing mean the authors omitted the patient from analysis? Otherwise no meaning of considering smoking and BMI as covariates.
  • Abbreviations should be elaborated under each table. Since the SEER does not corresponds to the SEER database, it would be better to describe what it is for.
  • The table preparation is very poor as the number within the parenthesis not described well. What is really the value? – mean or median
  • Hazard Ratio minimum and maximum values should be given.
  • Considering the action of GnRH agonist should take at least 6 months to show the efficacy, the reason for talking 3-months as cutoff point should be discussed.
  • Only antiandrogen – what drug?  

Minor comments:

The article should provide simple summary not repeating the authors instructions.

Author Response

Responses to Reviewer #3’s Comments

  1. As the years are limited and the prostate cancer survival is longer than the analyzed period(the median survival of PCa patients after ADT is 10-15 years), the authors should consider extending the period of analysis.

Response: We agree with the reviewer’s comment. Prostate cancer with lower tumor burden has a longterm survival. The shortest follow-up year of this cohort was only 1 year, which was not enough to evaluate the latest year cohort for dementia incidence. However, to consider the limitation of this study design with a population-based insurance dataset, we used an extended survival methodology to evaluate the association between ADT and dementia incidence. These mentioned limitations were also added in the limitations of the discussion section (page 19 line 303).

2.There is a clear definition for the cutoff in 2013. Again, neurological disorders (ND) are higher among patients who received second-generation antiandrogens. Considering the year 2013, it is hard to predict whether the study considers enzalutamide or apalutamide treatment. Again, part of ND is due to the antiandrogen itself and its action on androgen receptors present in neuronal cells.

Response: This study was not able to consider second generation of hormonal agents such as enzalutamide with ADT because most of the patients were before 2017. Further considerations for the association between ADT and dementia would be needed with the current second hormonal agent in ADT era. We have added this suggestion in the discussions section (page 19 lines 311-312).

3.The results are too brief, and the authors should elaborate the findings

Response: Due to the word limit of the journal guidelines, we simplified the important findings which might be too brief.

4.Did the missing mean the authors omitted the patient from analysis? Otherwise no meaning of considering smoking and BMI as covariates.

Response: The missing means that there is no point of considering smoking and BMI as covariates because they were insignificant in the univariate analysis.

5.Abbreviations should be elaborated under each table. Since the SEER does not corresponds to the SEER database, it would be better to describe what it is for.

Response: We have added abbreviations below each table (page 9,11, and 14).

6.The table preparation is very poor as the number within the parenthesis not described well. What is really the value? – mean or median

Response: Tables 1 and 3 are made more simplified. We have also added some scales to figure out the meaning of each number. Most of the variables were described in the form of the “means with SD” (page 8-14).

7.Hazard Ratio minimum and maximum values should be given.

Response: We have added the HR with minimum and maximum in the Results section according to the table 2 and tables 4-5 (page 4 line175 and page 5 lines 189-195).

8.Considering the action of GnRH agonist should take at least 6 months to show the efficacy, the reason for talking 3-months as cutoff point should be discussed.

Response: We stratified the ADT effect into 3 months and found that the cut-off of 3 months starting was appropriate for the incidence of dementia by the ADT. In addition, we defined the dementia that patients should take more than 3 months of dementia medication. Therefore, we chose the cut-off of 3-months in this study as a cut-off point.

9.Only antiandrogen – what drug? 

Response: Bicalutamide and flutamide are the only antiandrogen drug used for the “only antiandrogen” (page 3 line 108).

10.The article should provide simple summary not repeating the authors instructions.

Response: We have added simple summary before abstract (page 1 lines 15-22).
